# Bluetooth-Connected Pocket Spectrometer and Chemometrics for Olive Oil Applications

**DOI:** 10.3390/foods11152265

**Published:** 2022-07-29

**Authors:** Leonardo Ciaccheri, Barbara Adinolfi, Andrea Azelio Mencaglia, Anna Grazia Mignani

**Affiliations:** Istituto di Fisica Applicata “Nello Carrara” (IFAC-CNR), Via Madonna del Piano 10, 50019 Sesto Fiorentino, Italy; l.ciaccheri@ifac.cnr.it (L.C.); a.mencaglia@ifac.cnr.it (A.A.M.); a.g.mignani@ifac.cnr.it (A.G.M.)

**Keywords:** portable spectrometers, infrared spectroscopy, chemometrics, olive oil fatty acids

## Abstract

Unsaturated fatty acids are renowned for their beneficial effects on the cardiovascular system. The high content of unsaturated fatty acids is a benefit of vegetable fats and an important nutraceutical indicator. The ability to quickly check fat composition of an edible oil could be advantageous for both consumers and retailers. A Bluetooth-connected pocket spectrometer operating in NIR band was used for analyzing olive oils of different qualities. Reference data for fatty acid composition were obtained from a certified analytical laboratory. Chemometrics was used for processing data, and predictive models were created for determining saturated and unsaturated fatty acid content. The NIR spectrum also demonstrated good capability in classifying extra virgin and non-extra virgin olive oils. The pocket spectrometer used in this study has a relatively low cost, which makes it affordable for a wide class of users. Therefore, it may open the opportunity for quick and non-destructive testing of edible oil, which can be of interest for consumer, retailers, and for small/medium-size producers, which lack easy access to conventional analytics.

## 1. Introduction

There were about 3.8 billion smartphone users in the world in 2021 [1]. This ubiquitous penetration makes the smartphone the must-have Swiss-knife of the 21st century, thanks to the functions, sensors, and integrated gadgets it offers in addition to calling, texting, or surfing the web. The camera of the smartphone is a low-resource spectrometer, with an integral sensitivity in three bands somehow overlapping. Tablets offer similar performance, where the encumbrance due to their larger size is compensated by a wider screen that offers better data readability and easier typewriting. Consequently, smartphones and tablets can be conveniently used as photonic platforms for sensing and spectroscopy for a variety of applications, such as clinical diagnostics, health monitoring, biosensing, food analysis, safety, and allergen detection [2,3,4,5,6,7,8,9].

Bluetooth connectivity turns a smartphone into a technological octopus whose tentacles can interact with nearby electronic devices, while Wi-Fi connectivity and cloud-computing allow unprecedented functionalities, with limits dictated only by imagination and fantasy. A very convenient use of the smartphone is to exploit the Bluetooth connectivity for operating a miniaturized spectrometer. Photonic technologies are so advanced in miniaturization and cost that it is possible to integrate into a handheld device LEDs for illumination, spectrometer for detection, supply electronics, and a USB-rechargeable battery. Various models of such devices are already available [10,11,12,13,14].

This paper makes use of SCiO, a commercially available Bluetooth-connected spectrometer, and uses it for analyzing quickly and non-destructively important nutraceutical indicators of olive oil. This study was thought for:-advanced consumer applications: that is, instantly probing olive oil quality just shining light on the bottle,-importers, bottlers, and retailers, who wish to check the quality of the olive oil they bottle and sell, particularly the different batches they deal with,-small/medium-size producers in low-resource settings, for detecting the olive oil quality during production and storage, in addition to better product marketing. In fact, the measurement of nutraceutical and quality indicators is a precursor of generating a QR code, which points through the internet to the values of the measured parameters stored in a data warehouse. The possibility of creating a QR code for labeling leads to an innovative and modern marketing strategy, based on an interactive and rapid communication of oil quality to end consumers, giving to the product the chance of a digital life. Consumers who do not own a spectrometer but have a smartphone can retrieve product information, made available online by producers, simply scanning the QR code.

After being launched in 2014 employing a successful Kickstarter campaign, SCiO attracted much attention not only from consumers but also from research laboratories. Indeed, despite the low cost, it demonstrated good performance and functionalities, and demonstrated effectiveness for a variety of biomedical, industrial and food applications [15,16,17,18,19,20,21,22].

Near-infrared spectroscopy has already demonstrated its effectiveness in many fields, included foodstuff analysis [23]. It is more effective when used in combination with chemometrics, which is the science of extracting information from chemical systems by means of multivariate statistical analysis. Chemometrics is frequently used in NIR spectroscopy for analyzing multiple overlapped absorption bands.

Pocket spectrometers could provide a low-cost solution for putting analytics into the hands of consumers, small producers, and retailers. Wi-Fi connectivity of mobile phones is mandatory for such purpose. Because end-users are not usually trained in data processing, cloud computing is necessary for correctly processing data and obtaining the desired answer.

Thanks to our partnership with the SCiO producer in a joint Italian–Israelian research project, the raw spectroscopic data were accessible for our own custom processing and creation of original predictive models.

Distinguishing the olive oil type between extra-virgin olive oil (EVOO) or olive oil (OO) by means of a portable, easily operated device meets the consumer expectations. It also allows retailers to quickly check the product they sell, particularly when batches of olive oils are imported, blended, and locally bottled. Spectroscopy in near-infrared has been already used for this purpose. Many studies, however, were created using laboratory spectrophotometers that are not practical for low-resource users and on-field measurements [24,25,26,27,28,29].

Near-infrared spectroscopy, powered by chemometrics, also proved to be a successful technique for quantifying important nutraceutical indicators of olive oil, such as the content of saturated, mono-unsaturated, and poly-unsaturated fatty acids. However, in this case, conventional spectrophotometers were used, which are not suitable for consumer, retailers, or small-producer applications [30,31,32,33,34,35,36,37].

Olive oil is the primary source of fats in the Mediterranean diet and a key nutritional component responsible for most of the benefits of this type of diet. For these reasons, it is highly consumed and traded worldwide [38,39].

The chemical composition of olive oil is 98–99% a mixture of fatty acid esters, and the remaining 1–2% is a mixture of other compounds, typically hydrocarbons, tocopherols, phenols, alcohols, sterols, colored pigments such as carotenoids and chlorophylls, and secondary elements, such as aldehydes, ketones, and esters [40]. This chemical composition is largely dependent on the cultivar and the geographical origin, and, to a certain extent, olive ripeness and the extraction system.

Depending on the presence of double carbon-carbon bonds, fatty acids in olive oil are classified in three categories: saturated fatty acids (SFA) (palmitic acid, C16:0; heptadecanoic acid, C17:0; stearic acid, C18:0; arachidic acid, C20:0; behenic acid, C22:0), monounsaturated fatty acids, (MUFA) (palmitoleic acid, C16:1; heptadecenoic acid, C17:1; oleic acid, C18:1; eicosenoic acid; C20:1), and polyunsaturated fatty acids (PUFA) (linoleic acid, C18:2; and linolenic acid, C18:3) [41].

MUFA and PUFA play an essential role in the homeostasis and structure of the cell and the whole human body, being the main components of all biological membranes and source of energy stored in the triacylglycerols. Additionally, various metabolites of fatty acids serve as essential intracellular and extracellular lipid mediators [42,43]. While labelling of SFA is compulsory (Regulation EU n.1169/2011), the indication of MUFA and PUFA is a voluntary nutraceutical information that is gaining momentum as a marketing strategy to differentiate the products, particularly highlighting their nutritional advantages. As an example, Oleic acid is the famous omega-9 that lowers insulin levels and improves blood circulation [44,45]. Linoleic acid is an omega-3 essential acid that the human body cannot synthesize and must be up-taken from the diet. It is renowned for helping anti-atherosclerosis and anti-inflammatory mechanisms, and provides protection against neurodegenerative diseases [46,47,48,49].

The aim of this study is to employ SCiO spectra and chemometrics for distinguishing extra virgin from non-extra virgin olive oils and building predictive models for mono-unsaturated, poly-unsaturated, and saturated fatty acids.

## 2. Materials and Methods

### 2.1. Olive Oil Collection

The olive oil collection was made of 116 samples, produced in Italy, ranging from excellent to poor quality. This provided a representative training set for chemometric models:

-Extra virgin olive oils (EVOO) comprised 80 samples produced in two different years, thus considering the production variability. Some of them were provided by Fattoria Castel Ruggero Pellegrini, and some others were kindly supplied by Analytical Food, the official analytical laboratory hired for making reference analyses.-Lower quality edible oils (OO) comprised 36 samples; they were 24 samples of non-extra virgin olive oils, 8 samples of mixed pomace and olive oils, and 4 lampante oils (very low-quality virgin oil, not suitable for human consumption). All of them were supplied by Analytical Food or purchased on the market.

All the 116 oil samples were analyzed by Analytical Food for their acidic composition. Reference data were determined by gas chromatography following the Directive of the International Olive Council, COI-T.20-Doc. No 33-Rev.1-2017 [50]. Spectroscopic measurements were performed in parallel to the standard analytical measurements, to avoid alteration of samples caused by delays or wrong storage.

All chemical data were rounded according to their expanded uncertainty (±2 standard deviations). This prevented the analysis from being influenced by non-significant differences. The logarithm of mean concentration, expressed as fraction of unity, of every fatty acid is demonstrated in Figure 1. The total concentration of each category (SFA, MUFA, and PUFA) was calculated considering only fatty acids with mean concentration above 0.5%, because the others were known to have poor accuracy and added negligible corrections to the total of each category. This threshold is represented by a red line in Figure 1, where neglected fatty acids are represented by yellow bars. Therefore, SFA was calculated as the sum of palmitic and stearic (blue bars), MUFA as the sum of oleic and palmitoleic (red bars), and PUFA as the sum of linoleic and linolenic (green bars). Dynamic ranges (minimum ÷ maximum) were 12.5% ÷ 17.8% for SFA, 69.6% ÷ 80.3% for MUFA, and 5.2% ÷ 12.6% for PUFA, respectively.

### 2.2. Spectroscopic Measurements

SCiO is a battery-powered pocket-size device made of a bright near-infrared source, a micro-spectrometer for reflectance measurements in the NIR band, and a Bluetooth unit. The device has an array of 12 filtered sensors with different spectral sensitivity. The measured signals are sent to the driving device via Bluetooth connection and, from it, to SCiO web platform. There, the spectrum is reconstructed from the filter spectral sensitivities and the pattern of sensor outputs. Developers can download NIR reflectance spectra in the 740–1070 nm range at 1 nm step. Customers instead receive directly and instantly the required nutraceutical data on their mobile device, provided that a specific predictive model is available on the platform.

Figure 2 portrays the setup used at CNR laboratory for olive oil testing. A suitable holder was designed by CAD Rhinoceros^®^ and produced by a 3D-printer. It allowed for us to butt-couple SCiO to a 22 mm diameter glass vial in a repeatable manner and acquire the sample spectrum in trans-reflectance mode. In practice, this type of adapter simulated the use of the spectrometer with a glass bottle. An ASUS-ZENPAD tablet provided spectrometer control and connection with SCiO web platform during the laboratory testing. Each oil sample was measured five times, rotating the vial in front of the SCiO after each scan to average imperfections of the vial wall. For each oil sample, the average of the repeated measurements was used in data processing.

### 2.3. Data Analysis

Reflectance spectra were downloaded from SCiO website and converted to absorbance. Second derivative was applied for removing baseline fluctuations. Explorative analysis and data dimensionality reduction was then performed by means of principal component analysis (PCA), a widely used method for data compression and unsupervised feature extraction [51]. PCA searches for a set of orthogonal axes within the vectorial space spanned by input variables, maximizing the variance of projections along those axes. This allows for dimensionality reduction with minimal loss of information. PCA produces three sets of data. The loadings are the director cosines of principal axes in vector space and are used for detecting important variables. The scores are the projections of observations along those axes and are used for defining distances between observed samples and studying their clustering. The eigenvalues are the variances of projections along principal axes. Because the sum of all variances, the total variance, is invariant for PCA transformation, they allow for calculating the fraction of variance explained by each principal component (PC), hence its importance in data representation.

Chemometric models were trained with 78 samples (54 EVOO and 24 OO) and tested on the remaining 38 samples (26 EVOO and 12 OO). The training set was chosen using a Kennard–Stone (K-S) algorithm in PCA subspace to obtain uniformly spaced training spectra. The K-S choice was slightly adjusted to obtain a similar proportion of EVOO and OO in both subsets. Statistics of MUFA, PUFA, and SFA for calibration and validation set are provided in Table 1.

Discriminant analysis by means of quadratic discriminant analysis (QDA) has been performed for sample classification. QDA could not be applied directly to spectra, because it requires the number of observations in each class to be higher than that of variables. Therefore, PCA was used as preliminary data compression step. QDA was preferred to linear discriminant analysis because it does not assume all within-class covariance matrices being equal [52,53], which was not the case in our dataset.

QDA assumes that all classes follow a multivariate normal distribution, and scores were produced for each class and each point in PCA space, being proportional to logarithm of membership probabilities. Samples were then assigned to the most probable class according to their position in PCA space.

Regression models were created for predicting SFA, MUFA, and PUFA concentration by means of partial least square (PLS) regression [54]. PLS is a widely used tool when the predictor matrix contains many collinear variables, as it occurs in spectroscopy. In such cases, it avoids model instability and reduces the probability of overfitting. Like PCA, PLS works on the assumption that system response (the spectra) is driven by an unknown number of latent variables (chemical and/or physical) and project data on a sub-space whose axes are likely related with them. The fundamental difference is that PCA is an unsupervised method; it simply searches for spectral features having the stronger variance. PLS is a supervised method instead; it looks for features demonstrating better co-variance with target variable.

In practice, the PLS method allows us to predict the concentration of an analyte in *n*th sample by means of Equation (1):(1)$$\hat{y_{\;n}}=\;\;\;r_{\;0}+\;\overset{M}{\underset{m=1}{\sum }}r_{\;m}\;x_{\;n\;m}$$
where:*M*: number of spectral channels (discrete wavelengths),*r_0_* intercept of the linear model,*r_m_* regression coefficient corresponding to *m*th wavelength,*x_n m_* value of the spectrum of *n*th sample at *m*th wavelength.$\hat{y_{n}}$ predicted value of target variable for *n*th sample.


The regression models were rated according to their root mean square error of cross-validation (RMSECV), and root mean square error of prediction (RMSEP), the former measuring the accuracy of prediction on the calibration set and the latter on the validation set. For the validation set, two more parameters were calculated: the standard error of prediction (SEP) and the bias. Determination coefficients (*R*^2^) were calculated for comparing RMS errors with standard deviation of target variable. Bias is the mean of prediction deviations for validation set and accounts for systematic deviations between calibration and validation set. The ideal condition is when bias is zero. SEP is the RMS value of deviations corrected for bias. When bias is low, SEP is about equal to RMSEP. *R*^2^ coefficients are provided by Equation (2):(2)$$R^{2}=\;1-\;{\left(\frac{RMSE}{SD}\right)}^{2}$$
where *RMSE* is RMSECV for calibration set or RMSEP for validation set, while *SD* is the standard deviation of target variable within the considered data set. The ideal condition is when *RMSE* << *SD* and *R*^2^ tends to 1.

All data processing were made by means of *The Unscrambler^®^* 11 software package (Aspen Tech Bedford, Bedford, MA, USA, formerly CAMO Analytics, Oslo, Norway) [55]. Analysis of variance (ANOVA) was used for testing equality of means of single variables between EVOO and OO groups. Levene’s test was first used for testing hypothesis of homogeneous variance, and Welch’s correction was applied in case of significantly different variances at 5% level [56]. This analysis was performed by means of the open-source software JASP^®^ (The JASP team, University of Amsterdam, Amsterdam, Netherlands).

## 3. Results and Discussion

### 3.1. Classification of Olive Oil Type

A typical 2nd derivative spectrum of olive oil is demonstrated in Figure 3. According to Workman and Weyer [57], the main peak at 930 can be assigned to CH_2_ stretch, while the shoulder at 916 nm should be due to the CH_3_ stretch. The minor peak at 895 nm is likely due to stretching of olefinic = CH group. Workman and Weyer do not provide a precise wavelength for olefinic group in fatty acids, but they note that it occurs at shorter wavelength than saturated CH stretch. Thus, such assignment is reasonable.

The first two principal components (PCs) explained 73% of total variance and provided good separation of oil classes. Figure 4 portrays the PC2 vs. PC1 score plot, clearly indicating the difference between EVOO or OO groups. It also evident that OO class is more heterogeneous, demonstrating larger spread than EVOO. The strongest difference was along PC2, but also PC1 contributed to discrimination. ANOVA was used for testing the difference of class-means along PC axes. F values for PC1 and PC2, both calculated with Welch’s correction, were 6.8 and 75, respectively. The first was significant at a 5% level (*p* = 0.014) the second was significant at a 1% level (*p* < 0.001).

Loading spectrums of PC1 and PC2 are portrayed in Figure 5. PC2 has its strongest loadings placed at 892 nm (−) and 955 nm (+). The assignment of the latter is unclear, but the form is associated to absorption of unsaturated = CH groups. This suggests that discrimination ability is related to a different content of unsaturated fats between EVOO and OO. This is reasonable because oil is mainly made by fats, and fatty acids absorption dominates its NIR spectrum.

Indeed, OO indicated significantly higher mean PUFA content and lower mean MUFA content than OO, while SFA did not indicate significantly different group-means. Again, ANOVA was used for comparing variance of fatty-acid contents in EVOO and OO oils. SFA did not demonstrate any significant difference between EVOO and OO in either variance or mean (F = 0.01, *p* = 0.75). MUFA and PUFA instead portrayed highly significant differences in both. MUFA had F = 17, while PUFA had F = 32. Both results were significant at a 1% level (*p* < 0.001). These data suggested that a quantitative determination of fatty-acid contents from NIR spectra could be possible, and this was confirmed by the results of PLS regression.

PC1 and PC2 scores were fed as predictors, in a QDA classifier. Table 2 summarizes the confusion matrices for calibration and validation sets, respectively, where the following statistics were also provided:Accuracy: overall percentage of samples correctly classifiedSensitivity: true positive (EVOO) classification rate (ability to recognize an EVOO).Specificity: true negative (OO) classification rate (ability to reject an OO).

Three EVOOs and two OOs were misclassified in the training set, obtaining an accuracy of 94%, a sensitivity of 94%, and a specificity of 92%. Three false OOs were discovered in the validation set, thus providing 92% accuracy, 88% sensitivity, and 100% specificity.

Most misclassifications in either the training or validation set were false OO. This is likely due to the larger variance of OO class, resulting in a slower decrease of estimated probability density with the distance from group centroid. OOs are blends of refined olive or pomace oils and extra-virgin oil. The law set to 10% the minimum percentage of EVOO in the mixture, but its effective value, in addition to the quality of the oil used, is left to the producer. As consequence, OO is a rather heterogeneous class, and it can intersect the EVOO class.

Figure 6 portrays plots of discriminant scores for calibration (left) and validation (right) sets, respectively. Membership probability increases from left to right for the EVOO class and from bottom to top for OO class. The green line is the bisector of the 1st and 3rd quadrants (x = y) and represents the decision border. Note that EVOO scores span a much larger range than OO scores. This is due to more peaked density function of the EVOO group.

### 3.2. Prediction of Fatty Acid Content

Figure 7 portrays the predicted vs. reference plots for PUFA (left), MUFA (center), and SFA (right), respectively. Calibration and validation samples are indicated with blue and red symbols, respectively. Figure 8 portrays the regression coefficients for the three classes of fatty acids. Full regression statistics for calibration and validation set are provided in Table 3. PUFA demonstrates the best prediction, with a RMSEP of 0.39% and a R^2^ of 0.92. MUFA is only slightly worse, indicating a RMSEP od 0.74% and R^2^ of 0.90. These models can be considered satisfactory. SFA has an inferior performance. Its RMSEP is 0.55%, intermediate between the other two, but R^2^ is only 0.76 because its RMSEP is compared with a lower variance (see Equation (2)).

It is noteworthy the in both MUFA and PUFA coefficients spectra the band around 890 nm have strong weights. This band is related to CH stretch vibration of olefinic groups. These models indicate excellent results for predicting the concentration of PUFA, very good results for MUFA, and sufficient results for SFA. Indeed, SFA are harder to predict because they have lower variance than MUFA and PUFA. Standard deviation within calibration set is 1.1% for SFA, 1.5% for PUFA, and 2.2% for MUFA.

The ability of determining the SFA, MUFA, and PUFA is based on the balance between saturated and unsaturated carbon-carbon bonds, which modulate absorption of the olefinic = CH group. Such balance is primarily determined by those acids which have not only high concentration, but also a variance high enough for influencing the dataset. In our data set, the primary tradeoff between high-concentration acids was between oleic and linoleic, while palmitic was about the same for all samples. Therefore, the influence of palmitic on spectrum was relatively low. Conversely, stearic, the second-most abundant saturated acid in olive oil, was just 2% on average, so its influence was weak.

## 4. Conclusions

This study demonstrated a method for distinguishing extra-virgin from not extra-virgin olive oils by means of a single spectroscopic measurement in the NIR band and chemometric processing, with 92% accuracy. Data analysis suggested that such ability was correlated to different mean acidic composition of the two oil categories. This allowed us to determine contents of saturated, monounsaturated, and polyunsaturated fatty acids with uncertainty better than 1%.

These results demonstrated that the SCiO, a low-cost photonic device, can achieve multi-analysis of important nutraceutical indicators of olive oil, non-destructively, rapidly, and without chemicals. These interesting results open the possibility of developing a specific app for olive oil applications, making these analyses available to mobile device owners, such as consumers and olive oil producers.

Because all oils are primarily made of triglycerides, this app could potentially work on any edible oil. However, the present models have been calibrated on olive oils only. Further measurements on an expanded data set would be necessary for producing a reliable “all oils” app.

Moreover, we are planning further applications of SCiO in the olive oil sector, such as detection of cryptogamic diseases of the olive tree. The so-called peacock eye of the leaves is a deadly fungus responsible of *Cycloconium oleaginum* pathology that compromises the plant life, causing significant defoliation, which can lead to 80% decrease of fruit production. Preliminary reflectance measurements using SCiO of healthy and sick leaves has already demonstrated encouraging early results, suggesting the possibility of early diagnosis and intervention for promptly remediating the losses in oil production.

We take the opportunity of this publication to offer our collaboration worldwide to farmers, retailers, and consumers for investigating other food products, from the production to the transformation, final packaging, and distribution.

We have already 3D-printed several adapters to make SCiO compatible for measurements of solids, powders, and gels, and other custom adapters can be easily made for any type of food texture. We are keen to open our lab to anyone interested in testing their products, also offering our skills in chemometric data processing and data interpretation.

## Figures and Tables

**Figure 1 foods-11-02265-f001:**
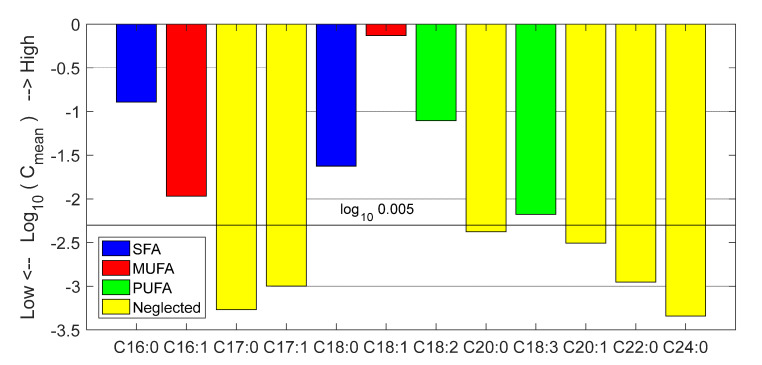
Logarithmic bar plot of analytically measured fatty acid mean contents of olive oils.

**Figure 2 foods-11-02265-f002:**
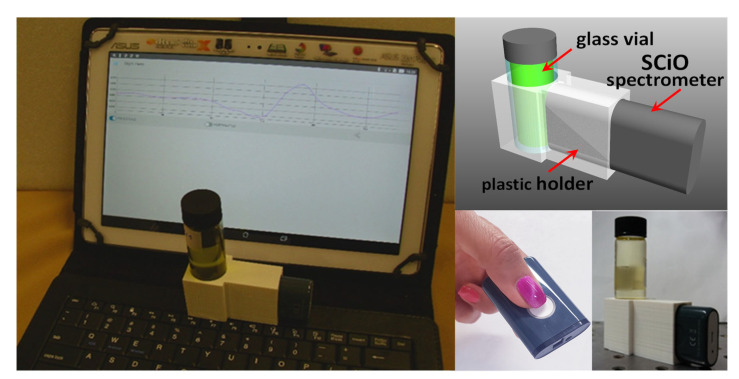
Photo of experimental setup (**left**). Rendering (**right-above**) and picture (**right-below**) of 3D-printed holder with SCiO handheld spectrometer and sample vial.

**Figure 3 foods-11-02265-f003:**
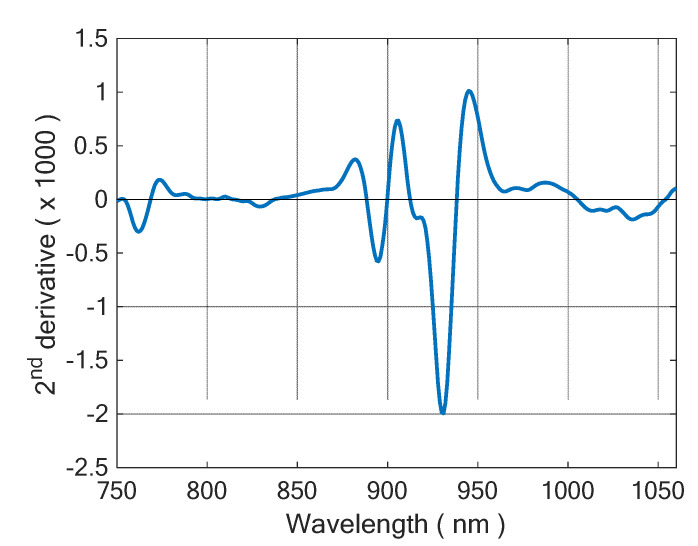
Typical 2nd derivative spectrum of olive oil.

**Figure 4 foods-11-02265-f004:**
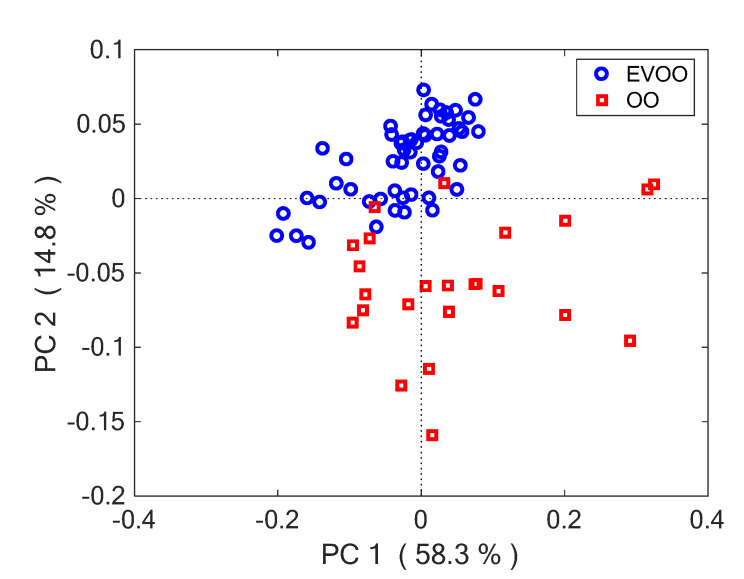
PC1-PC2 score plots, demonstrating the difference between EVOO and OO groups.

**Figure 5 foods-11-02265-f005:**
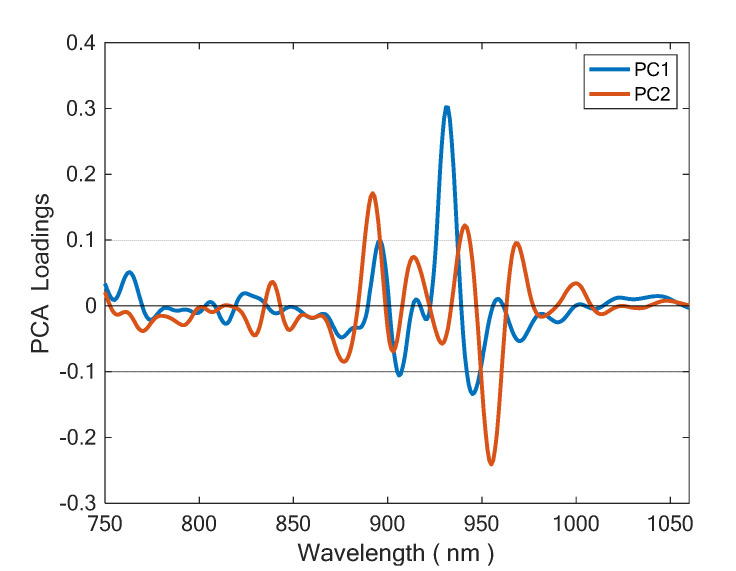
Loading spectra of PC1 and PC2, highlighting important variables.

**Figure 6 foods-11-02265-f006:**
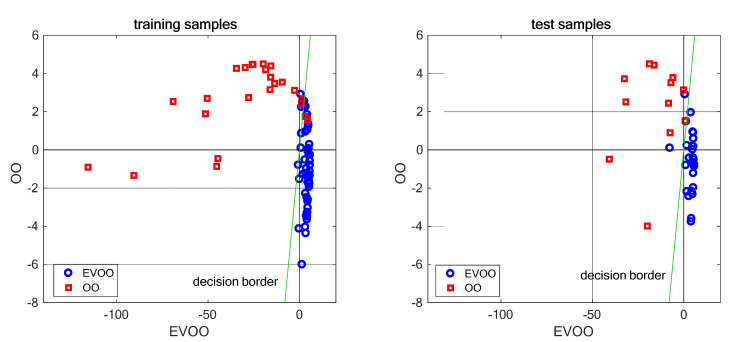
Plots of QDA discriminant scores for training (**left**) and test (**right**) samples.

**Figure 7 foods-11-02265-f007:**
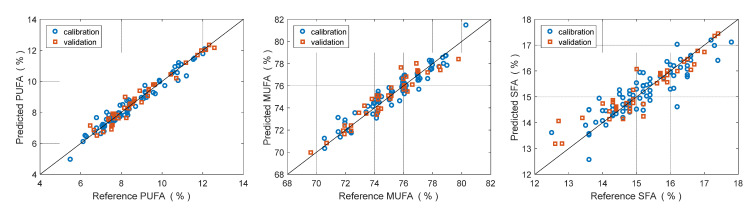
Predicted vs. Reference plots for concentration of PUFA (**left**), MUFA (**center**) and SFA (**right**). Blue circles: calibration. Red squares: validation.

**Figure 8 foods-11-02265-f008:**
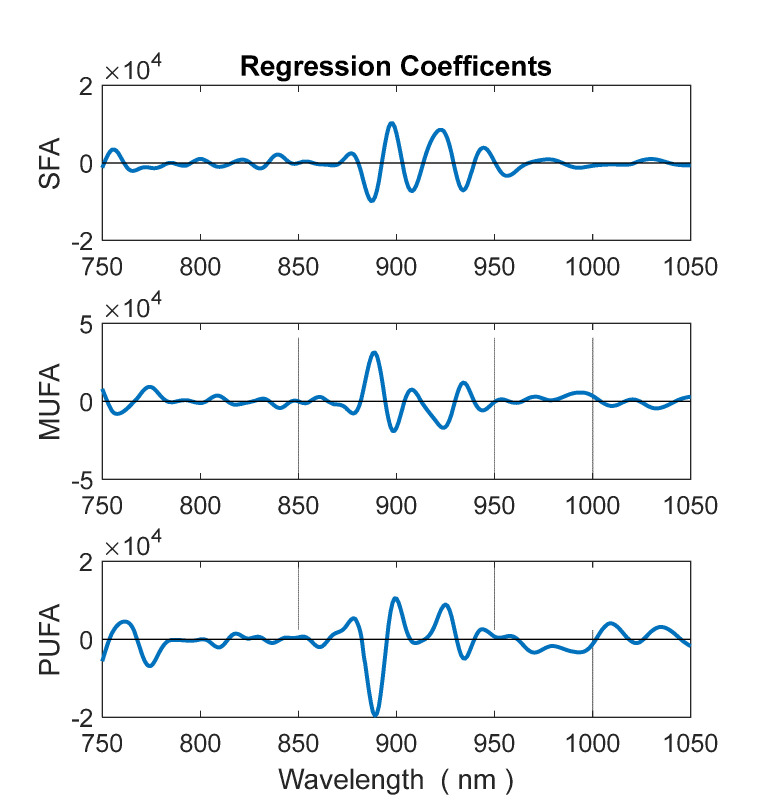
Regression coefficients of PLS models for PUFA (**bottom**), MUFA (**middle**), and SFA (**top**) contents.

**Table 1 foods-11-02265-t001:** Statistics of SFA, MUFA, and PUFA for calibration and validation set.

	Calibration Set		Validation Set	
	**Mean**	**SD**	**Mean**	**SD**
**SFA**	15.2%	1.1%	15.1%	1.1%
**MUFA**	75.1%	2.3%	75.7%	2.1%
**PUFA**	8.7%	1.6%	8.2%	1.5%

**Table 2 foods-11-02265-t002:** LDA classification rates and confusion matrices for training (left) and validation (right) sets.

Training			Validation		
**Accuracy 94%** **Sensitivity 94% Specificity 92%**			**Accuracy 92%** **Sensitivity 88% Specificity 100%**		
Predicted class	Actual class		Predicted class	Actual class	
	EVOO	Non-EVOO		EVOO	Non-EVOO
EVOO	51	2	EVOO	23	0
Non-EVOO	3	22	Non-EVOO	3	12

**Table 3 foods-11-02265-t003:** PLS regression statistics for prediction of PUFA, MUFA, and SFA contents. LV is the number of Latent Variables used in prediction.

		SFA	MUFA	PUFA
**Calibration**	LV	6	6	6
	RMSECV	0.51%	0.74%	0.37%
	*R^2^* (cal.)	0.780	0.895	0.950
**Validation**	RMSEP	0.55%	0.74%	0.39%
	SEP	0.55%	0.75%	0.39%
	Bias	0.06%	−0.11%	0.03%
	*R^2^* (val.)	0.764	0.872	0.921

## Data Availability

Not applicable.

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
