# Peer review of "Bluetooth-Connected Pocket Spectrometer and Chemometrics for Olive Oil Applications"

_foods, 2022, doi:10.3390/foods11152265_

Round 1

Reviewer 1 Report

The manuscript presents a method based on NIR coupled with a smartphone and chemometrics to predict saturated and unsaturated fatty acids content in olive oils and also classify them as extra-virgin.

 English should be revised.

 Abstract: at the end of the abstract the authors say, “This study opens the way towards quick and non-destructive testing of edible oil, which can be of interest for consumer, retailers, and for small/medium-size producers, which have not easy access to conventional analytical.” However, the consumers, retailers, and producers must have this equipment available correctly. So, in this way how about the costs of this pocket NIR (SCIO)? The device is really not expensive compared to other ones; however, they need to pay an annual signature. Maybe not very much accessible… may change the end of the abstract.

  Material and methods:

-          Please insert the Country origin of samples.

-          Please consider inserting some information regarding the fat acid sample preparation (esterification, trans-esterification…).

-          It would be interesting to make the holder project for 3D-print available as supplementary material.

 Results and discussion:

 -          Line 191: PCA for classification – consider explaining that PCA is for unsupervised classification. To analyze the oil class it would be necessary the LDA classification tool.

-          Equation 1 – vector should be in bold notation.

-          Concerning Fig 4, it would be interesting to include a plot (Fig 4B) showing PC1 x PC3 (2D scores plot).

-          Line 300 – Please explain better “R2 is only 0.76 because this variable has the lowest variance.”

-          Figure 7 – It would be interesting to try to assign the major analytical bands and relative peak positions for prominent near-infrared absorptions in the third overtone region. A suggestion is to see Figure 1 present in this monograph <https://www.metrohm.com/en/products/8/1085/81085026.html>

 Conclusions:

-          Second and fourth paragraphs are not conclusions concerning the presented work.

-          Consider re-write the conclusion highlighting the author's achievements.

Author Response

Please, see the attached files.

Reviewer 2 Report

I reviewed the manuscript entitled, Bluetooth-connected pocket spectrometer and chemometrics for olive oil applications. The manuscript has novelty and contributes to the field. Based on scientific soundness, the manuscript can be accepted for publication after addressing below suggestions

Line 18: …..to conventional analytical??? Is it analytical methods?. As such, it is incomplete

Introduction on chemometrics should be added in introduction section

Data analysis

Since this study highly focused on chemometrics, authors should describe the chemometric methods, such as PCA,

Authors should perform Discriminant analysis (DA)

Figure 4. why some of EVOO and OO are mixed in PCA and in Figure 5

Is this APP designed only for olive oil or other oils too?

Round 2

Reviewer 2 Report

Authors are now answered the questions/suggestions raised by me. In my opinion, this version can be accepted for publication in Foods.